

# Antiviral efficacy of short-hairpin RNAs and artificial microRNAs targeting foot-and-mouth disease virus

Anabella Currá, Marco Cacciabue, María José Gravisaco, Sebastián Asurmendi, Oscar Taboga and María I. Gismondi

Instituto de Agrobiotecnología y Biología Molecular (IABiMo), Instituto Nacional de Tecnología Agropecuaria (INTA), Consejo Nacional de Investigaciones Científicas y Técnicas (CONICET), Hurlingham, Buenos Aires, Argentina

Corresponding authors
Oscar Taboga,
taboga.oscaralberto@inta.gob.ar
María I. Gismondi,
gismondi.maria@inta.gob.ar

## ABSTRACT

RNA interference (RNAi) is a well-conserved mechanism in eukaryotic cells that directs post-transcriptional gene silencing through small RNA molecules. RNAi has been proposed as an alternative approach for rapid and specific control of viruses including foot-and-mouth disease virus (FMDV), the causative agent of a devastating animal disease with high economic impact. The aim of this work was to assess the antiviral activity of different small RNA shuttles targeting the FMDV RNA-dependent RNA polymerase coding sequence (3D). Three target sequences were predicted within 3D considering RNA accessibility as a major criterion. The silencing efficacy of short-hairpin RNAs (shRNAs) and artificial microRNAs (amiRNAs) targeting the selected sequences was confirmed in fluorescent reporter assays. Furthermore, BHK-21 cells transiently expressing shRNAs or amiRNAs proved 70 to >95% inhibition of FMDV growth. Interestingly, dual expression of amiRNAs did not improve FMDV silencing. Lastly, stable cell lines constitutively expressing amiRNAs were established and characterized in terms of antiviral activity against FMDV. As expected, viral replication in these cell lines was delayed. These results show that the target RNA-accessibility-guided approach for RNAi design rendered efficient amiRNAs that constrain FMDV replication. The application of amiRNAs to complement FMDV vaccination in specific epidemiological scenarios shall be explored further.

## INTRODUCTION

Foot-and-mouth disease virus (FMDV) is the causative agent of a highly devastating disease of cloven-hoofed animals leading to enormous economic losses. FMDV belongs to the Aphthovirus genus in the *Picornaviridae* family. There are seven FMDV serotypes (A, O, C, Asia1, SAT1, SAT2, SAT3) and several subtypes.

The viral particle is ∼25–30 nm in diameter and consists of a single stranded (+)RNA molecule surrounded by an icosahedric capsid composed by the four structural viral proteins VP1 to VP4. Viral infection begins with the interaction of the FMDV capsid with cellular receptors ($\alpha_V$ integrins in animal hosts, additionally heparansulfate and a newly

identified receptor in cultured cells) (*Dicara et al., 2008*; *Lawrence et al., 2016*). Receptor-mediated endocytosis guides the viral particles to the cytoplasm, where the whole replication cycle takes place. Acidification of endosomes drives disassembly of the viral capsid and release of the FMDV genome into the cytoplasm, where it is immediately translated in a cap-independent manner to produce the viral polyprotein. Co- and post-translational processing of the FMDV polyprotein originates viral protein precursors and ultimately the mature structural and non-structural proteins (*Grubman & Baxt, 2004*). Finally, capsids are assembled and newly synthesized genomes are encapsidated and released from the cell in a lytic way. The replication cycle is fast, taking approximately 6–8 h in cell culture.

During replication, viral RNA molecules are synthesized by the viral RNA-dependent RNA polymerase (RdRp) 3Dpol. As other viral RdRps, the enzyme lacks proofreading activity, which contributes to the generation of mutated FMDV genomes in every replication cycle. Hence, in an infected host (cell or animal) the virus appears as a swarm of different but closely related viral variants which are called the viral quasispecies (*Domingo et al., 2003*; *Domingo, Sheldon & Perales, 2012*; *Andino & Domingo, 2015*). The quasispecies nature of FMDV facilitates its rapid adaptation to unfavorable environments.

Following current standards of the World Organisation for Animal Health (OIE), countries are divided according to their foot-and-mouth disease (FMD) status in FMD-free countries with or without vaccination (almost all Americas, Europe and Oceania) and FMD-endemic countries (parts of Africa and Asia). Current approved FMD vaccines are inactivated viruses formulated with oil-based adjuvants. Although they are effective, inactivated vaccines show a number of disadvantages including lack of cross-protection between serotypes and even subtypes, inability to induce long-lasting immunity (there is a need of re-vaccination to maintain protection) and risk of incomplete inactivation during manufacturing. Moreover, current vaccines take ∼7 days to induce protection (*Golde et al., 2005*). Thus, there is a need of developing new tools against FMDV that may complement vaccination and reduce the risk of infection especially during outbreaks.

RNA interference (RNAi) is a well-conserved mechanism in eukaryotic cells that directs post-transcriptional gene silencing (*Fischer, 2015*). RNAi is mediated through small RNA molecules that guide ribonucleprotein complexes to their cognate mRNAs and lead to their degradation and/or inhibition of translation. There are many types of small RNAs, including microRNAs (miRNAs) and short interfering RNAs (siRNAs). The former are endogenous molecules encoded in the cellular genome that regulate gene expression. In animals, biogenesis of miRNAs involves RNA polymerase II-mediated transcription of a primary miRNA (pri-miRNA), which is further processed by the RNAse III enzyme Drosha to pre-miRNA hairpins. After export from the nucleus, pre-miRNAs are processed by Dicer to produce mature 21–24 nt miRNAs that exert their function in the cytoplasm of the cell. In turn, siRNAs are exogenous molecules that may be introduced into the cell either as mature siRNAs or as DNA precursors by transfection or by virus-mediated transduction (*Castanotto & Rossi, 2009*). Importantly, the degree of complementarity between the small RNA and the target RNA defines the mechanism of post-transcriptional gene silencing: while perfect base complementarity mainly drives target RNA degradation by endonucleolytic cleavage, partial complementarity can lead to translational repression.

In animals, the canonical siRNA pathway involves perfect siRNA:target RNA recognition, whereas binding of most miRNAs demands base pairing of the 'seed' region (mostly located at nucleotides 2–8 of the mature miRNA), allowing for imperfect target RNA recognition (reviewed in (*Carthew & Sontheimer, 2009*)).

The use of RNAi as an antiviral tool has been explored against different viruses (reviewed in (*Qureshi et al., 2018*; *Shah & Schaffer, 2011*)). Indeed, RNAi mediators such as double-stranded or single-stranded siRNAs and plasmid or virus-encoded short hairpin RNAs (shRNAs) have been evaluated in different experimental settings against animal viruses including FMDV (*Chen et al., 2004*). Genetically modified animals expressing shRNAs targeting FMDV have also been developed with promising results (*Jiao et al., 2013*; *Hu et al., 2015*; *Deng et al., 2017*). Of note, shRNAs are potent RNAi mediators that usually are expressed at very high levels from RNA polymerase III promoters; however, this can potentially lead to saturation of the cellular RNAi processing machinery. To overcome this drawback, artificial miRNAs (i.e., miRNAs rationally designed to target defined sequences) have been favored as RNAi shuttles since they exhibit increased safety with comparable efficacy (*Boudreau, Martins & Davidson, 2009*). The efficacy of artificial miRNAs (amiRNAs) against FMDV has been investigated to a lesser extent (*Du et al., 2011*; *Gismondi et al., 2014*; *Basagoudanavar et al., 2019*). In this work, we further assess the efficacy of amiRNAs against FMDV and shRNAs targeting the same viral regions in susceptible cells.

## MATERIALS AND METHODS

### Cells and viruses

Baby hamster kidney (BHK-21 clone 13, ATCC CCL10) cells were obtained from the American Type Culture Collection and maintained in Dulbecco's modified Eagle's medium (DMEM, Life Technologies, Grand Island, NY, USA) supplemented with 10% fetal bovine serum (FBS) and antibiotics (Gibco-BRL/Invitrogen, Carlsbad, CA, USA) at 37 °C with 5% $CO_2$. Stably transformed cell lines were grown as described in *Gismondi et al. (2014)*.

FMDV A/Arg/01 isolate A01L (GenBank accession number KY404934) was obtained from the National Institute for Animal Health (SENASA, Argentina). All experiments were conducted using fourth cell passages in BHK-21 cells. Experiments involving active FMDV were performed in the BSL-4 OIE facility at the IVIT (INTA-CONICET; Buenos Aires, Argentina).

### Plaque assays and virus titration

Plaque assays were performed as previously described in *Garcia-Nunez et al. (2010)*. Briefly, cells were infected with serial dilutions of virus and incubated for 48 h at 37 °C with 5% $CO_2$. Monolayers were fixed with 4% formaldehyde and stained with crystal violet, and the number and area of individual plaques ($n \geq 18$) were determined by using ViralPlaque (*Cacciabue, Curra & Gismondi, 2019*). Assays were performed in duplicate.

Virus titers were determined in BHK-21 cells as described in *Gismondi et al. (2014)*.

**Table 1  Accessibility parameters of selected target sequences.**

| Predicted target sequence (5′–3′)[a] | Mature sRNA sequence (5′–3′)[b] | RNAxs ranking position[c] | Accessible window (starting position; length) | $\Delta G_{total}$ (kcal/mol)[d] | sRNA ID[e] |
|---|---|---|---|---|---|
| CGUUUACGAAGCAAUCAAA | **C**UUUGAUUGCUUCGUAAACG**C** | 2; 3 | 13; 9 | −22.2 | 290 |
| GGAGAACAGAGAAUACAAA | UAUUCUCUGUUCUC**CAUGAGC**[*] | 1; 6 | 17; 5 | −22 | 444 |
| CUCAGGCCCCACUUUAAAU | AUUUAAAGUGGGGCCUGAG**AG** | 5 | 11; 16 | −29.6 | 1055 |
| | AUAGUCCAUGUGGAAGUGUCU | | 18; 5 | | 1162 |
| | AAAUGUACUGCGCGUGGAGAC | | | | Neg |

**Notes.**

[a] 19 nt target sequences were predicted with RNAxs software.

[b] 21 nt target sequences were defined according to the rules described in the Results section. The nucleotides incorporated to the 19 nt predicted sequence are shown in bold.

[c] Ranking position of 19 nt predicted target sequences comprised within the 21 nt predicted sRNA sequences.

[d] $\Delta G_{total}$ was calculated with OligoWalk software, implemented in RNA structure package.

[e] sRNAs 1162 and Neg were used as positive and negative controls, respectively.

[*] sRNA 444 slightly differed from the predicted target sequence to favor a less structured RNA conformation at the target 3′ end (Fig. S1)

## Prediction of RNAi target sequences

Target sequences within FMDV 3Dpol coding region (nucleotides 6681-8093 of FMDV A01L) were predicted with RNAxs software (*Tafer et al., 2008*). This program is freely available at http://rna.tbi.univie.ac.at/cgi-bin/RNAxs/RNAxs.cgi. It relies on several siRNA design criteria, namely siRNA strand selection (using both sequence and energy-based rules), self-folding (the minimal free folding-energy of the guide strand), free-end (number of paired nucleotides among the first four at the 5′ end and the 3′ end of the guide strand) and most importantly accessibility (probability that a region of predefined length is free of base pairing in thermodynamic equilibrium). The program (accessed by October 2014) was run with default values for siRNA. The output file of RNAxs includes a number of putative target sequences of 19 nt. To adapt the prediction software to 21 nt target sequences, 2 additional nucleotides were incorporated to each designed small RNA (see Results). Lastly, small RNAs targeting 3Dpol sequence 1162-1182 were also used since this region showed high flexibility in a local SHAPE-directed RNA structure obtained at our laboratory (Fig. S1 and unpublished results). Moreover, it was previously demonstrated that this stretch of FMDV O/CHA/99 can be successfully targeted by RNAi (*Gu et al., 2014*). In addition, a plasmid encoding a control sequence not expected to target any mammalian mRNA, EGFP or FMDV was used as a negative control.

## Plasmids
### shRNA encoding plasmids

RNA polymerase III U6 promoter including a BbsI site at the 3′ end was amplified by PCR and cloned within plasmid TOPO pCR2.1 (Invitrogen, Carlsbad, USA) to create plasmid pTOPO-U6. Complementary single-stranded DNA oligonucleotides encoding shRNA against predicted target sequences (Table 1; Table S1) were synthesized (Macrogen, Seoul, Korea), annealed and cloned into the BbsI restriction site in pTOPO-U6. The resulting plasmids were named pshRNA$_{FMDV}$, where FMDV is the target sequence ID.

### Pre-amiRNA encoding plasmids

Individual pre-amiRNA coding sequences were cloned into pcDNA® 6.2-GW/miR vector (BLOCK-iT Pol II miR RNAi expression vector kit, Invitrogen, Carlsbad, USA) as described in *Gismondi et al. (2014)*. The resulting plasmids were named pamiRNA$_{FMDV}$, where FMDV is the target sequence ID.

Dual amiRNA$_{FMDV}$-expressing plasmids were assembled from individual pamiRNA$_{FMDV}$ according to the manufacturer's instructions (BLOCK-iT Pol II miR RNAi expression vector kit, Invitrogen, Carlsbad, USA). Briefly, donor pamiRNA$_{FMDV}$ DNA was digested with BamHI and XhoI and the purified pre-amiRNA$_{FMDV}$ encoding sequence was subcloned into acceptor pamiRNA$_{FMDV}$ previously digested with BglII and XhoI enzymes.

### Enhanced green fluorescent protein (EGFP) reporter plasmids

A reporter plasmid encoding the EGFP gene fused to the complete FMDV 3Dpol-coding sequence (pEGFP.3D) was constructed. To this end, the EGFP coding sequence was amplified by PCR using oligonucleotides EGFPfor and EGFPrev (Table S1) and cloned between KpnI and EcoRI restriction sites in pcDNA3 vector (Invitrogen, Carlsbad, USA). In addition, the FMDV A01L 3Dpol-coding sequence was amplified by PCR and cloned immediately downstream of the EGFP termination codon.

All recombinant vectors were confirmed by automated sequencing.

## Transfection and electroporation

Pre-amiRNA$_{FMDV}$ and shRNA$_{FMDV}$-encoding plasmids were transfected into 95% confluent BHK-21 cells seeded in 48-well culture plates using PolyAr reagent (School of Biochemistry and Pharmacy, University of Buenos Aires), according to manufacturer's instructions. Briefly, cells were washed with serum-free DMEM, and incubated with 125 µl of transfection mixture containing 900 ng DNA, 1 µl PolyAr and DMEM for 4 h at 37 °C and 5% $CO_2$. Transfected cells were washed three times with sterile 1X PBS and incubated for additional time in DMEM 10% FBS at 37 °C and 5% $CO_2$.

When indicated, $1.3 \times 10^6$ BHK-21 cells were electroporated with 6 µg DNA using GenePulser XCell (BioRad, Hercules, CA, USA). Electroporation conditions included 1 pulse at 280 V for 25 ms in four mm-cuvettes. Electroporated cells were seeded in 48-well culture plates and incubated in DMEM 10% FBS at 37 °C and 5% $CO_2$ for 18 h prior to FMDV infection.

## Reporter assays

Plasmid pEGFP.3D was co-transfected with either pshRNA$_{FMDV}$ or pamiRNA$_{FMDV}$ at a 1:3 molar ratio using PolyAr as described above. At 24 h posttransfection, EGFP expression was evaluated by fluorescence microscopy in a Zeiss microscope. In addition, transfected cells were washed 2 times with 1X PBS, trypsinized and fixed in 1% formaldehyde for 20 min on ice. After 2 washes in 1X PBS 2 mM EDTA, EGFP expression was quantified by flow cytometry. A BD FACSCalibur and FlowJo™ Software (Becton, Dickinson and Company; 2019) were used to measure frequency and median fluorescence intensity of EGFP+ cells. Of the 10,000 events evaluated per sample, only those with the forward scatter and side scatter properties of viable cells were used in measurements of EGFP fluorescence intensity.

### Establishment of transgenic cell lines

At 24 h post-transfection with pamiRNA$_{FMDV}$, cells were subcultured in DMEM medium containing 10% FBS and 7 μg/ml blasticidin S (Invitrogen, Carlsbad, USA). Transgenic polyclonal cell lines were established by consecutive passages in DMEM 10% FBS and 7 μg/ml blasticidin S.

Polyclonal cell lines were further cloned by limiting dilution as described in *Gismondi et al. (2014)*. Transgenesis was confirmed by PCR using oligonucleotides miRNAseq for and miRNAseq rev (Table S1).

### Antiviral activity of small RNAs against FMDV

Electroporated cells were infected for 60 min at a multiplicity of infection (moi) of 3–5. Cells were washed with 1X PBS pH 5.0 on ice to inactivate unabsorbed virus. After restitution of physiological pH, cells were incubated in DMEM 2% FBS and 25 mM HEPES pH 7.5 at 37 °C in a 5% $CO_2$ atmosphere. At indicated times post-infection, cells were lysed by three consecutive freeze-thaw cycles and the amount of viral particles was measured by the TCID$_{50}$ method as mentioned above. Percentage of inhibition was calculated from the following equation: $P = ((A - B) \times 100)/A$, where $A$ was viral titer of cell lines expressing miRneg infected with FMDV A/Arg/01, and B was viral titer of cell lines expressing amiRNA$_{FMDV}$ infected with FMDV A/Arg/01. Experiments were performed in duplicate.

Stable cell lines expressing amiRNA$_{FMDV}$ were infected at an moi of 0.001 and incubated in DMEM 2% FBS and 25 mM HEPES pH 7.5 at 37 °C and 5% $CO_2$. At different times post-infection, the virus present in supernatants was titrated by the TCID$_{50}$ method.

### Statistical analysis

Student's *t* test was used to compare mean viral titers in different samples and mean plaque areas. Differences in median fluorescence intensity in cells transfected with different plasmids were compared using Kruskal-Wallis test. A *p* value <0.05 was considered as statistically significant.

## RESULTS

### Selection of target sequences

Target sequences were predicted within FMDV A01L 3Dpol-coding sequence with RNAxs software (*Tafer et al., 2008*), which applies several prediction parameters including RNA accessibility. A total number of 178 target sequences were predicted along 3D (Fig. 1, blue areas; Table S2), with top 50 predicted targets concentrated in discrete regions (Fig. 1, red areas).

Next, an additional screening of the target sequences ranking in the top positions of the RNAxs output was performed to take into account the prediction rules proposed by *Low et al. (2012)*. Firstly, 19 nt target sequences were extended to 21 nt in order to adapt RNAxs prediction to amiRNA-compatible length. Then, the resulting target sequences were analyzed in terms of the accessibility window proposed by Low et al., i.e., sequences showing a less structured stretch downstream of the 21 nt target region were favored. Lastly, the total free energy change ($\Delta G_{total}$) of target:siRNA duplex formation was calculated

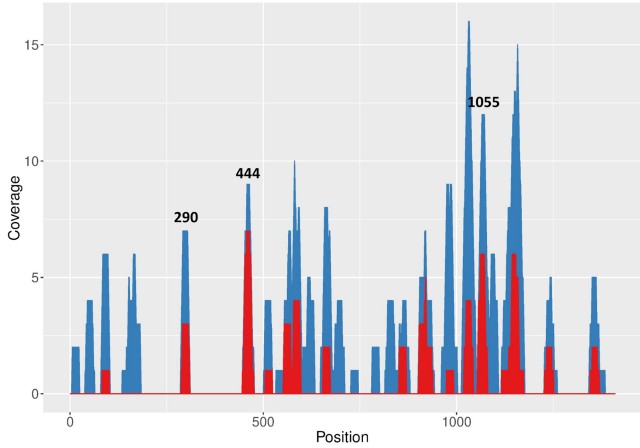

**Figure 1** **RNAxs results of target prediction within FMDV 3D region.** Data are represented as the number of predicted target sequences (coverage) per 3D nucleotide. Total number of predicted sequences (blue areas) and sequences corresponding to the top 50 hits in the RNAxs ranking (red areas) are shown. Target sequences selected for further study are indicated at the top of the corresponding peaks.

with OligoWalk program (*Reuter & Mathews, 2010*) (Table 1). After applying these rules, sequences 444–464, 290–310 and 1055–1075 were selected (termed 444; 290 and 1055; Fig. 1, Table 1 and Fig. S2). The selected target sequences ranked in the first positions of the RNAxs output. Indeed, in some cases the 21 nt target sequences included more than one 19 nt predicted target sequence (Table 1). Of note, sequences 290 and 444 displayed a $\Delta G_{total}$ value higher than the one recommended by Low et al. (−25 kcal/mol). Regarding the accessibility window, no target sequence showed the expected accessible 13 nt window starting at position 14 (Fig. S1A). However, all sequences were located in the region of high linear correlation between RNA structure and shRNA efficacy described by these authors (Fig. S1B).

Finally, conservation of the selected target sequences among prototypic strains of the South American serotype pool was evaluated (Fig. S1C). Sequences 290, 444 and 1055 showed some polymorphisms between serotypes, which were distributed mainly outside the seed region.

## Silencing efficacy of shRNAs and amiRNAs targeting selected FMDV sequences

A reporter assay was conducted to test the silencing efficacy of different RNAi mediators targeting the selected sequences. To this end, a plasmid encoding EGFP fused to the 3Dpol-coding sequence of FMDV A/Arg/01 was co-transfected with plasmids encoding either shRNAs or pre-amiRNAs targeting each individual FMDV sequence. Short hairpin RNAs targeting sequences 290, 444 and 1055 induced a significant reduction of EGFP expression in co-transfected cells, as determined by fluorescence microscopy and flow cytometry (Fig. 2A and Fig. S2A). In turn, amiRNAs targeting single FMDV sequences also exhibited significant silencing activity (Fig. 2B and Fig. S2B).

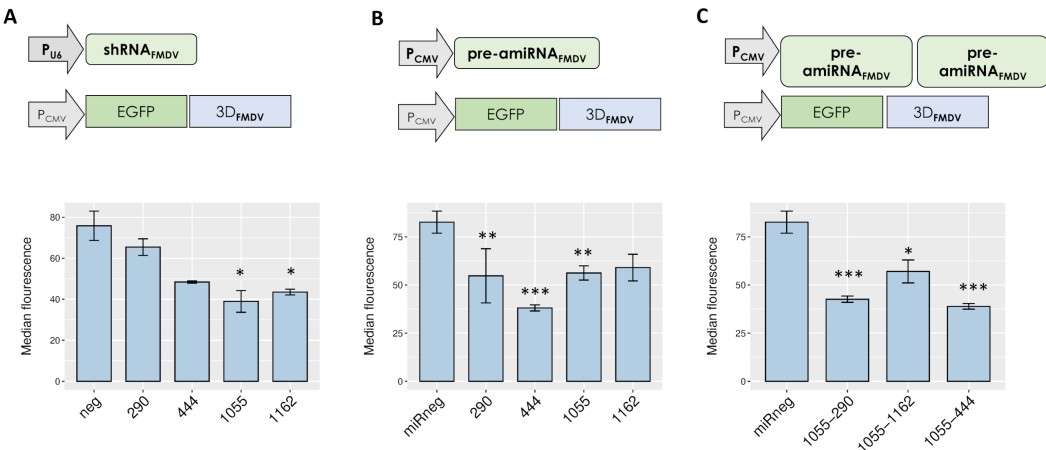

**Figure 2  Silencing activity of small RNAs directed against FMDV 3D sequences.** Co-transfected cells were trypsinized and EGFP expression was analyzed by flow cytometry at 24 hpi as described in Material and Methods. Cells were co-transfected with pEGFP. 3D and shRNA_FMDV (A), pre-amiRNA_FMDV (B) or dual pre-amiRNA_FMDV (C) expressing plasmids. $*p < 0.05$; $**p < 0.01$; $***p < 0.001$.

To evaluate the impact of dual amiRNA_FMDV expression on silencing activity in reporter assays, pre-amiRNA_290, pre-amiRNA_444 or pre-amiRNA_1162 were subcloned in pamiRNA_1055 to obtain bicistronic expression constructs and the resulting plasmids were co-transfected with pEGFP.3D in BHK-21 cells. As shown in Fig. 2C, dual expression of amiRNAs did not enhance silencing of the reporter mRNA.

## Antiviral activity of shRNA_FMDV and amiRNA_FMDV

Next, we evaluated the antiviral activity of shRNAs and amiRNAs against FMDV in cultured cells. To this end, BHK-21 cells were electroporated with the small RNA-expressing plasmids, and after 16 h the cells were infected with FMDV at high multiplicity of infection. Viral titers were determined at different times post infection to assess the effect of small RNAs on viral growth. As listed in Table 2, inhibition of viral growth was evidenced at 5 h post-infection (hpi) in cells expressing small RNAs targeting regions 290, 444 and 1055. Interestingly, shRNA_290 induced a 2-fold reduction of FMDV replication, whereas shRNA_444 and shRNA_1055 induced a more pronounced decrease in viral titers at this time point. Conversely, there was a higher inhibition of viral growth in cells expressing amiRNA_290 (>10-fold) than in cells electroporated with pamiRNA_444 and pamiRNA_1055. Again, dual expression of amiRNAs targeting different FMDV sequences did not improve the silencing activity of each individual amiRNA (Table 2). Of note, the silencing efficacy of small RNAs determined previously in our reporter assays was confirmed in the context of viral replication except for shRNA_1162, which did not affect FMDV replication significantly.

## Stable expression of amiRNAs against FMDV

Electroporation of DNA is a well-established method that yields a high frequency of transformants (*Potter, 2003*); however, transfection efficiency is not 100%. We hypothesized

**Table 2    Antiviral activity against FMDV exerted by amiRNAs and shRNAs at 5 hpi.**

| Target region | % inhibition | |
|---|---|---|
| | amiRNA | shRNA |
| 290 | 96 | 48 |
| 444 | 90 | 87 |
| 1055 | 71 | 77 |
| 1162 | 60 | 10 |
| 1055-290 | 90 | nd |
| 1055-444 | 74 | nd |
| 1055-1162 | 61 | nd |

**Notes.**
nd, not done.

that electroporation of plasmids expressing small RNAs could lead to a population of non-transfected cells that, when infected, would contribute to FMDV replication. To circumvent this possibility, stably-transformed cell lines expressing amiRNA$_{290}$ and amiRNA$_{1055}$ were established and characterized in terms of antiviral efficacy. In this case, only amiRNAs were evaluated since they exhibit less toxicity than shRNAs (*Boudreau, Monteys & Davidson, 2008*).

Pre-amiRNA$_{FMDV}$ expressing plasmids were stably transfected in BHK-21 cells and transgenic polyclonal cell lines were selected by blasticidin resistance. Mature amiRNA expression was confirmed by RT-stem loop qPCR (Fig. S4A). After several passages, cells were infected with FMDV A/Arg/01 and cultures were monitored to evaluate the development of cytopathic effect. At 48 hpi, control cell monolayers (non-transfected BHK-21 cells and cells stably transfected with a non-specific amiRNA) were completely detached due to FMDV infection. In contrast, infection of cell lines stably expressing amiRNA$_{290}$ and amiRNA$_{1055}$ caused partial cytopathic effect, with a significant proportion of cells still adherent to the culture surface (Fig. 3A). This effect was also evidenced in lysis plaques assays, where FMDV infection rendered less plaques of smaller dimensions in cell lines expressing amiRNA$_{290}$ and amiRNA$_{1055}$ than in control cells (Fig. 3B). However, viral titers determined at different times post-infection in supernatants of amiRNA$_{290}$ and amiRNA$_{1055}$ polyclonal cells did not differ significantly from control cells except for amiRNA$_{290}$ cells at 18 hpi (Fig. 3C). Together, these results suggest a transient amiRNA-mediated constraint on FMDV replication in transgenic cell lines.

Next, polyclonal cell lines expressing amiRNA$_{290}$ and amiRNA$_{1055}$ were cloned by limiting dilution to consider the potential variability in transgene expression associated with multiple transgene insertion events during the establishment and selection of stable transformants. The resulting cell lines were infected with FMDV A/Arg/01 and their ability to control viral growth was assessed by plaque assay (data not shown). Cloned cell lines showing less or smaller plaques than control cells (BHK-21 and miRneg-expressing cells) were selected for further analysis. Selected cell lines ($n = 4$ for amiRNA$_{290}$ and $n = 3$ for amiRNA$_{1055}$) were infected with FMDV A/Arg/01 and viral growth was determined by viral titration in supernatants at different times post-infection. As shown in Fig. 4, viral
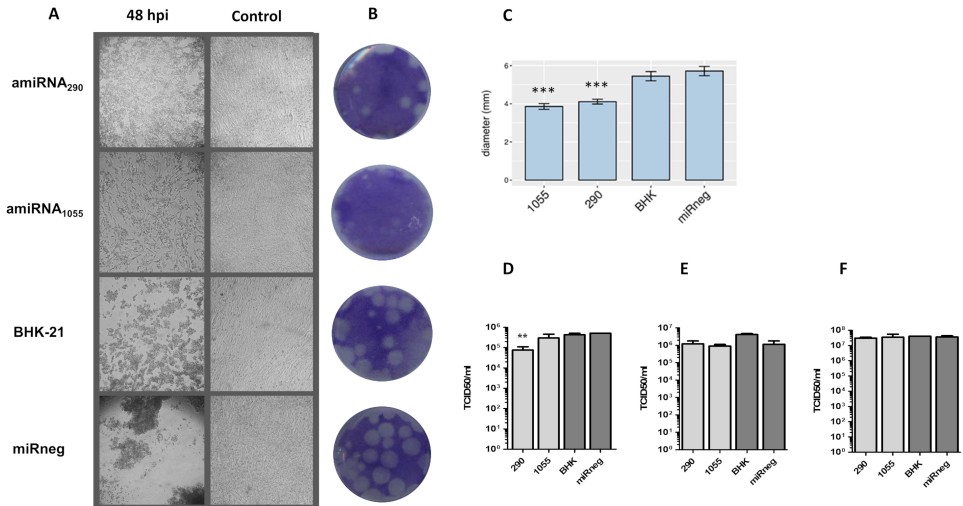

**Figure 3** **Antiviral effect of amiRNAsFMDV stably expressed in BHK-21 cells.** (A) Cytopathic effect observed in BHK-21 cell lines expressing amiRNAFMDV and control group (cell lines expressing miRneg) at 48 h post-infection. Cellular detachment, rounding, and destruction were more severe in the control group than in the experimental group (Magnification 100 ×). (B) Morphology and (C) dimensions of lysis plaques produced by FMDV in amiRNAFMDV-expressing cell lines and control cells. FMDV infection produced less plaques of smaller dimensions in cell lines expressing amiRNA than in control cells. *** $p <$ 0.0001. (D–F) Viral titers in supernatants of infected cells were determined by the $TCID_{50}$ method at 18 hpi (D), 24 hpi (E) or 48 hpi (F). ** $p < 0.01$.

growth was delayed in the majority of the cloned cell lines, supporting the potential use of amiRNAs as antivirals. However, at 48 hpi, the viral titers in supernatants of infected transgenic cells did not differ from control cells. Moreover, direct sequencing of a 250–300 nt region encompassing the target site did not show any mutation that could account for virus escape (Fig. S5).

## DISCUSSION

To develop an antiviral strategy against FMDV based on RNAi, the 3Dpol-coding sequence of FMDV A/Arg/01 was used to predict target sites. This region of the viral genome is highly conserved within FMDV serotypes. Moreover, the essential role played by the protein product (3Dpol) in the viral replication cycle makes it an attractive target for RNA interference. As shown in the reporter assays, the predicted targets were silenced efficiently independently of the RNAi mediator used, supporting the feasibility of the algorithm applied for target selection. Our results are in accordance with previous data indicating that the 3D region of the FMDV genome is a suitable target for RNAi (*Pengyan et al., 2008*; *Du et al., 2011*; *Gu et al., 2014*; *Basagoudanavar et al., 2019*) and they represent a step forward in the identification of new target sequences within FMDV, specifically in serotype A, that had been less explored in previous works.

It is well known that target accessibility may limit RNAi efficacy (*Westerhout, 2005*; *Tafer et al., 2008*; *Low et al., 2012*; *Gismondi et al., 2014*). In a paper on HIV inhibition by RNAi, *Low et al. (2012)* analyzed the impact of target RNA structure and target:siRNA duplex

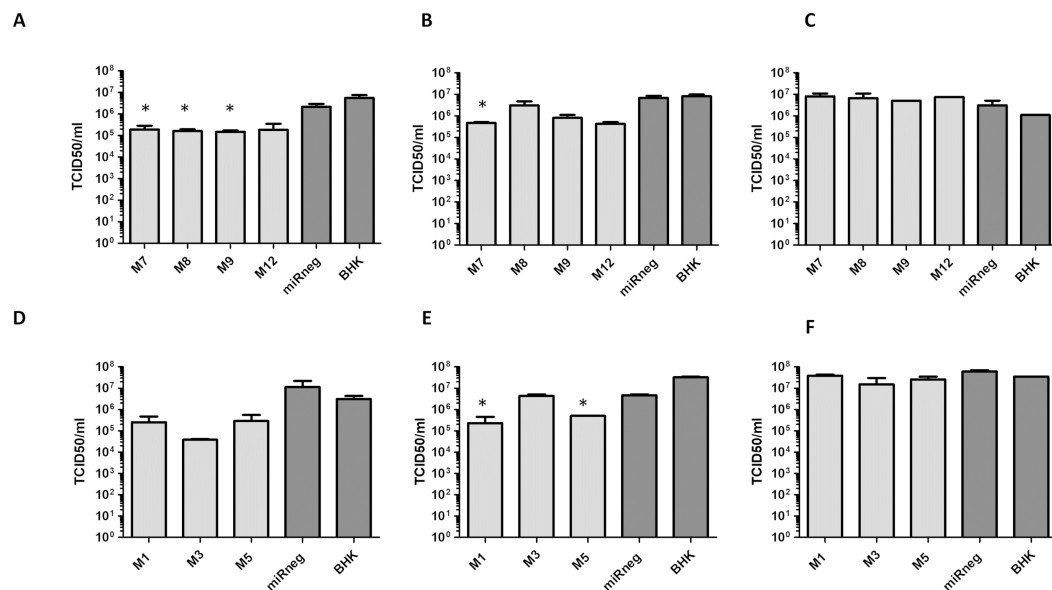

**Figure 4** **Antiviral activity of cloned amiRNA$_{290}$ (A–C) and amiRNA$_{1055}$ (D–F) cell lines.** Cell lines were obtained from polyclonal cell lines by limiting dilution as described in Material and Methods. Cells were infected with FMDV A/Arg/01 at a low moi and viral titers in supernatants at 18 hpi (A, D), 24 hpi (B, E) or 48 hpi (C, F) were determined by end-point dilution. * $p < 0.05$ (as compared to miRneg cells, Student's $t$-test).

thermodynamics on shRNA efficacy. They found a maximal negative correlation for a 13-nt window that starts at position 14 of the target RNA and extends over 7 nucleotides downstream of the 'seed region' binding site. In the context of RNAi, a negative correlation implies that maximal shRNA inhibition is achieved for less structured target RNAs. Moreover, $\Delta G_{total}$ of target:siRNA duplex formation strongly correlated with shRNA inhibition. Thus, in this work RNA accessibility was used as a central criterion for target selection.

The available programs for prediction of RNAi targets can be classified as first- and second-generation algorithms. The latter are based not only on a number of selection criteria but also they were optimized to include results from real RNAi experiments to validate their predictions (*Griger & Tisminetzky, 1984*; *Devaney et al., 1988*; *Filhol et al., 2012*). In the present work we used a first-generation RNAi target selection program but we also included the experimental data derived from the work by *Low et al. (2012)* to predict RNAi targets. Interestingly, none of the selected regions had been targeted previously in other similar studies with other FMDV serotypes (*Chen et al., 2004*; *Kahana et al., 2004*; *Mohapatra et al., 2005*; *Du et al., 2011*). Furthermore, our analysis with RNAxs did not retrieve the targets selected by other authors within the 15 best ranking sequences or it did even not predict them at all. This may indicate that, despite sequence conservation between FMDV serotypes in the 3D region, accessibility of specific target sequences may not be equivalent between serotypes due to structural constraints in the context of the whole genome, as we previously reported.
Indeed, conservation of the target sequence among different viral strains should be considered during the development of RNAi-based antiviral tools (*Mcintyre et al., 2009*). In this work, selected sequences 290, 444 and 1055 showed partial conservation among South American serotypes; however, the sequences used for small RNA design were the most frequent at polymorphic sites (Fig. S1C). Interestingly, other stretches within 3D showing 100% conservation were not predicted in the top-ranking target sites by RNAxs software, suggesting these sequences may not be accessible. Initial reports on determinants of miRNA:mRNA recognition highlighted the existence of two categories of target sites. 5′ dominant sites, which are strongly complementary to the miRNA 5′ end, need little or no pairing of the miRNA 3′ end to achieve silencing. On the other hand, miRNAs with partial complementarity at the 5′ end require strong 3′ compensatory pairing for function (*Brennecke et al., 2005*; *Bartel, 2009*). Thus, the existence of polymorphisms in the target sequence should not preclude miRNA efficacy against different FMDV serotypes. Moreover, potentially reduced RNAi due to weak amiRNA:target RNA pairing at polymorphic sites could be circumvented by co-expression of multiple amiRNAs targeting different target sequences of the viral genome.

Interestingly, the shRNA directed against sequence 1162 was effective in the reporter assay and the amiRNA trended to perform in a similar way, although with no statistical significance. Contrarily, $shRNA_{1162}$ did not affect FMDV replication significantly, whereas $amiRNA_{1162}$ showed moderate antiviral activity. This target region was selected as a putative positive control, since the same sequence of FMDV serotype O/CHA/99 (Fig. S3) had been efficiently silenced by an shRNA in a previous paper (*Gu et al., 2014*). Noteworthy, as mentioned before, this region was not predicted as a suitable RNAi target by RNAxs software. It may be argued that $shRNA_{1162}$ did not work due to structural constraints of the target region in the context of the whole viral genome, which were not present in the reporter mRNA. However, the results obtained with $amiRNA_{1162}$ indicate that target site accessibility should not be limiting $shRNA_{1162}$ efficacy. Alternatively, it is possible that this target region is not completely accessible to the mature small RNAs, leaving only some nucleotides available for small RNA:target recognition. In this case, pairing of the seed region of $amiR_{1162}$ would be sufficient for activation of the RNAi machinery, whereas incomplete pairing between $shRNA_{1162}$ and its target RNA would not lead to RNA silencing. In addition, since it has been shown that siRNA derived from shRNA precursors are more heterogeneous in cleavage sites and length than small RNAs originating from miRNA precursors (*Maczuga et al., 2013*), differences in shRNA and pre-amiRNA structures and processing that may lead to slightly different mature small RNAs with singular silencing efficacy cannot be ruled out.

Short hairpin RNAs are highly transcribed from RNA polymerase III promoters such as the U6 promoter used in this study. In addition, it has been demonstrated that shRNAs are more potent than amiRNAs (*Boudreau, Monteys & Davidson, 2008*) when they are expressed from the same promoter. However, shRNAs exhibit higher cellular toxicity than amiRNAs since the high levels of shRNA precursors produced in a cell may saturate the endogenous miRNA processing machinery (*Boudreau, Martins & Davidson, 2009*). In our reporter assays, shRNAs and amiRNAs were transcribed from different promoters (U6 and

CMV, for RNA polymerases III and II, respectively) and we did not detect any difference in cell viability between cells transfected with pshRNA$_{FMDV}$ and cells transfected with pamiRNA$_{FMDV}$ as determined by flow cytometry.

Our results show that amiRNAs induce a significant inhibition of viral replication in cultured cells; however, silencing of FMDV RNA is not complete leading to sustained viral replication. Notably, amiRNAs were poorly expressed in the majority of cloned cell lines as compared to polyclonal cells (Fig. S4B). Thus, the lack of sustained viral inhibition may be associated with insufficient amiRNA expression. However, a ∼24 h delay in viral replication has been also reported by *De Los Santos et al. (2005)*, *Chen et al. (2004)* and *Lv et al. (2009)* using shRNAs or siRNAs. Conversely, *Kahana et al. (2004)* showed a sustained inhibition of FMDV infection in BHK-21 cells transfected with siRNAs against FMDV; however, the authors only analyzed viral replication for 24 h. In turn, by using an siRNA transfected in BHK-21 cells, *Mohapatra et al. (2005)* reported a reduction of viral titer of >99% at 24 hpi and >87% at 48 hpi. The transitory character of viral inhibition may be explained by several factors. First of all, it should be kept in mind that in the infected cell the FMDV RNA is associated with multiple cellular and viral proteins during the processes of genome translation and replication. Thus, target sequences should not be expected to be continuously accessible for the silencing complexes during the whole FMDV replication cycle. In addition, it is well known that FMDV proteases induce a rapid shut-off of cellular transcription and cap-dependent translation (*Kim et al., 2008*; *Qureshi, Thakur & Kumar, 2013*). Given that amiRNAs are expressed from the strong CMV promoter, it is expected that pre-amiRNA transcription is also inhibited in infected cells in the context of viral replication. Thus, the incomplete inhibition of FMDV replication by amiRNAs may be explained by an FMDV-mediated reduction in transgene expression.

Other authors have demonstrated that simultaneous expression of multiple small RNAs directed against different regions of the same target RNA can increase silencing efficacy (*Kahana et al., 2004*; *Brake et al., 2008*; *Liu et al., 2008*; *Kim et al., 2010*; *Saha et al., 2016*). This approach becomes particularly relevant when the target RNA is a viral genome, since multiple small RNAs acting on the same genome reduce the chance of emergence of escape mutants thus increasing safety. However, in this work dual amiRNA expression did not enhance silencing of the reporter mRNA or the complete FMDV genome as compared to individual amiRNA expression. Similar results have been reported by other authors using amiRNAs against FMDV (*Basagoudanavar et al., 2019*). Noteworthy, amiRNA expression did not differ significantly in cells stably transfected with monocistronic or bicistronic vectors (Fig. S4A). Further research is needed to determine whether it is due to structural constraints of the target RNA or to the intrinsic regulation of the RNAi machinery.

## CONCLUSIONS

In light of the results obtained, we conclude that the selection algorithm proved to be highly accurate in the identification of target sequences. FMDV replication may be transiently controlled by amiRNAs targeting 3D. Ongoing work in our laboratory is aimed at finding new targets in other genomic regions of FMDV to develop additional amiRNAs that could

potentiate the antiviral effects observed in the present work. Our results highlight the potential application of amiRNAs as antiviral tools which may complement vaccination, specifically to prevent the spread of the disease during outbreaks in endemic or non-endemic countries.

# ACKNOWLEDGEMENTS

We thank María José Mónaco and Osvaldo Zábal for technical assistance and Juan Manuel Schammas and Alfredo Perea for their support during our work in the BSL4-OIE facilities at the Institute of Virology and Technical Innovations (INTA-CONICET).

## Funding

This work was supported by the Instituto Nacional de Tecnología Agropecuaria (PNBIO 1131034) and by Agencia Nacional de Promoción Científica y Tecnológica (PICT 2017-2581). The funders had no role in study design, data collection and analysis, decision to publish, or preparation of the manuscript.

## Grant Disclosures

The following grant information was disclosed by the authors:
Instituto Nacional de Tecnología Agropecuaria: PNBIO 1131034.
Agencia Nacional de Promoción Científica y Tecnológica: PICT 2017-2581.

## Competing Interests

The authors declare there are no competing interests.

## Author Contributions

- Anabella Currá conceived and designed the experiments, performed the experiments, analyzed the data, prepared figures and/or tables, authored or reviewed drafts of the paper, and approved the final draft.
- Marco Cacciabue performed the experiments, prepared figures and/or tables, authored or reviewed drafts of the paper, and approved the final draft.
- María José Gravisaco performed the experiments, analyzed the data, authored or reviewed drafts of the paper, and approved the final draft.
- Sebastián Asurmendi and Oscar Taboga conceived and designed the experiments, authored or reviewed drafts of the paper, and approved the final draft.
- María I. Gismondi conceived and designed the experiments, analyzed the data, authored or reviewed drafts of the paper, and approved the final draft.

## DNA Deposition

The following information was supplied regarding the deposition of DNA sequences:
FMDV A01L sequence was used for target selection (GenBank: KY404934). The sequences corresponding to Fig. S5 are available at GenBank: MW423371 to MW423381.

## Data Availability

Raw data are available in the Supplementary File.

## Supplemental Information

Supplemental information for this article can be found online at http://dx.doi.org/10.7717/peerj.11227#supplemental-information.

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
