# Peer review of "Antiviral efficacy of short-hairpin RNAs and artificial microRNAs targeting foot-and-mouth disease virus"

_PeerJ, doi:10.7717/peerj.11227_

## Round 0.1 · original submission · Major Revisions

Your manuscript has been carefully reviewed by three experts. I agree with them that it deserves to be published. However, there are interesting and important points that need to be addressed in order to accept your manuscript for publication. A relevant point that I would like to know more abut is how the targets have been selected and if previous reports on the variability in these regions have been taking into account. Please include the suggested experiments and all the changes the reviewers have asked for as they will improve your manuscript. Also, I would like to receive a point-by-point answer to all reviewer criticisms and suggestions.

Reviewer 1 ·

Basic reporting

No comment

Experimental design

No comment

Validity of the findings

No comment

Additional comments

This is a study on RNAi strategies against the FMDV pathogen. Similar studies have been published several years ago, but this study has a few somewhat novel items (e.g. amiRNAs) that make it worth reporting, but several items need to be improved. A quite minimal set of experiments were performed, thus some additional work can be done.

The focus is on accessibility of the RNA targets, but this means that another critical variable is ignored: conservation of the target sequences among different FMDV isolates. What is the value of a good inhibitor if it only attacks the lab strain used? Pioneering studies about this have been done for HIV. At least the authors should indicate if the chosen/optimal targets are conserved and refer to previous work on this variable.

As a main focus is on target RNA accessibility, it seems important to refer to some of the key initial reports (PMCID: PMC1934999). In fact, the same group showed that RNA structure changes can lead to viral escape (PMCID: PMC548362). The latter finding indicates that sequencing of just the target sequence does not suffice when one studies viral escape! The authors did not detect viral escape mutations, but perhaps one first has to passage the virus.

Too many terms are used, sRNA shRNA siRNA amiRNA. Perhaps sRNA (was not explained) can be removed for simplicity.

58: remove very

60: OIE?

84: exportation > export

267: chained is quite a popular word. I would like to see details as the inability to see better inhibition with the combination is strange and likely due to an expression issue. I cannot judge that as the basic information on the chaining strategy was not provided. Were 2 expression units coupled, are thereby repeats introduced that may affect vector stability etc. But besides this basic information, it may be required to perform RNA expression studies to demonstrate that both inhibitors are expressed.

Table 2: decimals suggest an accuracy that is probably not there.

Reviewer 2 ·

Basic reporting

no comment

Experimental design

no comment

Validity of the findings

nothing to add

Additional comments

The paper "Antiviral efficacy of short-hairpin RNAs……" by Anabella Currá et al., outlines their study on testing FMDV silencing by examining inhibition by shRNA as well as miRNA, for achieving better control of the disease, predominantly in FMDV endemic regions. In this study, two RNAi approaches were undertaken. The first, a transient suppression assay, using shRNAs, and miRNAs expression vectors, targeted three 3D sites: 444, 290, and 1055. The three shRNA as well as three amiRNA vectors were cotransfected with a pEGFP3D reporter plasmid for examining the level of GFP expression. The results, as shown in Fig. 2, demonstrate a maximum reduction of ~40% (compared to ~80% GFP expression level in the control assay), where shRNA and amiRNA targeted at the 444 position showed to be the most effective silencing site.
There are several issues concerning the assay. First, why is the pEGFP3D built as it is? Why not the other way around? Why wasn't the 3D gene cloned downstream to the GFP sequence and in the same reading frame? By doing so, the identification of silencing would be much more pronounced and precise. A second comment is, why not try to cotransfect two or three of the mono construct and the GFP reporter vector instead of using the double expression vectors? Using these methodologies might give much better results.
As for the second method of testing suppression activity, namely infecting stable RNAi BHK clones, can the authors comment on whether they were looking to find the highest expressed clone for each construct before the suppression assays?
One more issue is the lack of discussion on the authors' vision of how RNAi could be applied to combat the disease and how the present study advances their ultimate goal.
Otherwise, The paper is well written, employing the appropriate controls, even though the results they obtained were less convincing by comparing with other reports, and the authors should reflect this piece of information.
In summary, the manuscript is worth publishing after responding to the issues raised in this review

Reviewer 3 ·

Basic reporting

This study seems to be a continuous work of Gismond et al (J. Virol. Methods 2014).
Anti-viral effect of shRNA and amiRNA is interesting and authors need to address following points to strength their findings.

Major points
1. In Fig.3., the quantitative data of FMDV-RNA in each experiment groups may clarify the effect of amiRNA.
2. Expression level of amiRNAs in each cells should be better to show in Fig.3 and Fig.4 to evaluate their efficacy.
3. Information about statistical analysis is better to be provided in Fig.4.

Minor points
1. Please provide references for sentence (line 61-62).
2. Please add reference (line 351).
3. Please provide magnification (Fig.3A).

Experimental design

Please see basic report

Validity of the findings

Please see basic report

Additional comments

Please see basic report

---

## Round 0.2 · Minor Revisions

Please address the four minor issues noted by the reviewer in the annotated manuscript and modifiy the manuscript accordingly prior to acceptance. Thanks

Reviewer 2 ·

Basic reporting

No comment

Experimental design

No comments

Validity of the findings

No comments

Additional comments

See attached file.

Annotated reviews are not available for download in order to protect the identity of reviewers who chose to remain anonymous.

---

## Round 0.3 · accepted · Accept

I believe your manuscript is now ready for publication. Congratulations and thank you very much for your patience.